# Post-Operative Complications Do Not Influence Time to Adjuvant Treatment in Breast Cancer Patients Undergoing Implant-Based Reconstructions: Pre-Pectoral Versus Sub-Pectoral

**DOI:** 10.3390/cancers18010109

**Published:** 2025-12-29

**Authors:** Gianluca Vanni, Marco Pellicciaro, Marco Materazzo, Alice Bertolo, Amir Sadri, Elisa Campanella, Denisa Eskiu, Ilaria Portarena, Benedetto Longo, Valerio Cervelli, Oreste Claudio Buonomo

**Affiliations:** 1Breast Unit Policlinico Tor Vergata, Department of Surgical Science, Tor Vergata University, Viale Oxford 81, 00133 Rome, Italy; 2PhD Program in Applied Medical-Surgical Sciences, Department of Surgical Science, Tor Vergata University, 00133 Rome, Italy; 3Plastic Surgery, Great Ormond Hospital for Children NHS Foundation Trust, London WC1N 3JH, UK; 4Faculty of Medicine, Università Cattolica Nostra Signora Del Buon Consiglio, 1001 Tirana, Albania; d.eskiu@prof.unizkm.al; 5Department of Oncology, Tor Vergata Hospital, 00133 Rome, Italy; 6Plastic and Reconstructive Surgery, Department of Surgical Science, Tor Vergata University, 00133 Rome, Italy; 7Department of Health Science, University of Basilicata, Via Nazario Sauro, 85, 85100 Potenza, Italy

**Keywords:** breast cancer, implant-based reconstruction, time to adjuvant treatment, pre-pectoral breast reconstruction, post-operative complications

## Abstract

Mastectomy is undoubtedly a highly invasive procedure and has a serious psychological and physical impact on patients. The introduction of pre-pectoral reconstruction has led to an increase in immediate breast reconstructions, improving patients’ quality of life and reducing surgical morbidity, as there is no need to create a sub-pectoral pocket. Although these advantages are well established, some authors have reported a higher risk of implant-related complications. These complications could potentially delay the initiation of adjuvant treatments, which is well known to negatively affect oncologic outcomes. In our study, we compared pre-pectoral and sub-pectoral reconstructions and found no statistically significant differences in the time to adjuvant treatment. Therefore, given its lower surgical impact, the pre-pectoral approach should be considered the preferred option whenever feasible in patients undergoing mastectomy for breast cancer.

## 1. Introduction

Despite improvements in oncoplastic techniques that have expanded the indications for breast-conserving surgery, approximately 30% of breast cancer patients still undergo mastectomy [1]. Over the years, significant changes have occurred in mastectomy procedures [2]. When oncologically feasible, preservation of the nipple areola complex (NAC) and as much skin as possible is now standard practice, allowing for immediate breast reconstruction [2,3]. In parallel with the rise of sparing mastectomies, the traditional two-stage reconstruction, consisting of the placement of a sub-muscular tissue expander followed by permanent implant exchange, has increasingly been replaced by immediate pre-pectoral breast reconstruction, performed with or without or without mesh coverage [4]. It was initially adopted because it enabled immediate single-stage reconstruction, with remarkable benefits in terms of both quality of life and psychological well-being for patients [3]. Compared to traditional sub-muscular reconstruction, the pre-pectoral technique minimizes surgical trauma by avoiding muscle dissection, as the implant is placed directly above the pectoralis major [4]. This approach results in reduced post-operative pain, improved esthetic outcomes, and faster functional recovery, allowing patients to resume daily activities more quickly [3,4,5,6]. Being a less invasive approach, many surgeons have also adopted the pre-pectoral reconstruction technique in two-stage procedures, also involving tissue expanders [6,7,8].

Despite the advantages, some studies have reported a higher risk of complications, including implant removal, seroma formation incidence, and delayed wound healing [9,10]. Such events may prolong surgical recovery and consequently delay the starting of adjuvant treatments, potentially impacting oncologic outcomes [11].

Importantly, while complication rates are frequently reported, their clinical relevance in terms of the delayed initiation of adjuvant therapies has been insufficiently investigated, despite the well-known impact of treatment timing on oncologic outcomes [11]. Therefore, there is a need for comparative analyses that simultaneously evaluate the reconstruction plane, staging strategy, and post-operative outcomes in order to identify factors associated with increased complication risks and treatment delays.

The aim of our retrospective study was to compare the complication rates among patients undergoing immediate or two-stage pre-pectoral and sub-pectoral breast reconstruction and to evaluate potential predictive factors associated with longer intervals between surgery and adjuvant treatments.

## 2. Materials and Methods

This retrospective, single-center study included all patients who underwent mastectomy at the Breast Unit of Policlinico Tor Vergata, Rome, between January 2014 and March 2024, with a minimum follow-up of one year. Patients who underwent autologous breast reconstruction or who did not receive any breast reconstruction were excluded from the analysis.

Patient and surgical data, including age, the presence of comorbidities (specifically type II diabetes mellitus), and smoking status, were retrieved from clinical records. Patients who had undergone previous radiotherapy were reported and analyzed. The timing of radiotherapy was recorded and categorized as less than or greater than 10 years. Mastectomy flap thickness was reported according to the Rancati classification based on the evaluation of mammographic imaging [12]. Information regarding the type of mastectomy was obtained from surgical procedure reports. Mastectomy types categorized as “conserving” encompassed procedures including nipple–areola complex-sparing, skin-sparing, or skin-reducing mastectomies. Surgical time was defined as the total operating room occupancy, including patient positioning, anesthesia, and the surgical procedure itself. Indications for mastectomy were also retrieved from clinical notes and included in the analysis. The volume of the removed breast was retrieved from the final histopathological report and recorded in cubic centimeters, cc. The implant volume was retrieved from the surgical operative report and defined as the volume of the definitive implant in direct-to-implant reconstructions or the volume of the tissue expander placed in two-stage reconstructions.

Patients who received adjuvant radiotherapy after mastectomy were also reported. According to our institutional policy, these patients undergo radiotherapy within 3 months after surgery.

Surgical procedures were performed by an oncoplastic breast surgeon. Following pre-operative patient marking, mastectomy was carried out through meticulous dissection within the plane of the subcutaneous fascia. The dissection was performed with preservation of the dermal blood supply by carefully separating the avascular plane between the subcutaneous fat and the glandular tissue. Scissors were used for the dissection of the superficial plane, and monopolar diathermy was employed for the dissection of the pectoral plane. The breast was dissected from the pectoralis major muscle fascia using monopolar diathermy.

The type of reconstruction, either with a silicone breast prosthesis or a tissue expander, was determined from surgical notes. For direct-to-implant (DTI) reconstructions, a sizer was used to assess the appropriate volume and achieve symmetry. The placement of the prosthesis was recorded as sub-pectoral (Figure 1), pre-pectoral with or without mesh (Figure 2 and Figure 3) or dual-plane. For the purpose of this study, patients were categorized into two distinct groups according to the reconstruction type: pre-pectoral or sub-pectoral. Patients who underwent dual-plane reconstruction were excluded from the primary comparison between these two groups.

Surgical complications occurring within one year after surgery were reported in clinical records and were classified as follows: delayed wound healing, defined as full-thickness wound separation; bleeding, defined as post-operative anemia requiring re-operation for hemostasis or blood transfusion; and seroma and hematoma, defined as fluid or clot accumulation necessitating aspiration or drainage. One-year implant loss, defined as prosthesis removal due to surgical complications such as infection, exposure, and/or dislocation, was reported and analyzed between groups. Post-operative complications were classified according to the Clavien–Dindo classification. Complications graded I–II were considered minor and did not require a surgical procedure in the operating room, whereas grades III–V were considered major complications [13]. Patients without complications, showing complete wound healing and full resolution of the surgical condition, were considered eligible to start adjuvant treatment. The time interval between surgery and resolution of the surgical condition was defined as the time interval to adjuvant treatment. A 60-day cut-off was used to identify delays in adjuvant treatment.

The study was registered and approved by the Ethics Committee of Lazio Area 2 (approval number: 62.25CET2 PTV). No potentially identifiable images or patient data are presented in this study. All procedures involving human participants were conducted in accordance with the ethical standards of the 1964 Helsinki Declaration. Formal patient consent was not required due to the retrospective nature of the study.

### Statistical Analysis

Data were collected in a Microsoft Excel database (Microsoft 2023, Version 16.78) and subsequently analyzed using SPSS version 23.0 (IBM Co., Armonk, NY, USA). The chi-squared test or analysis of variance (ANOVA) was applied to compare patient characteristics, with data presented as median values and corresponding ranges. The chi-squared test was also used to compare complication rates between different reconstruction types, with results reported as absolute numbers and relative percentages. Univariate logistic regression analyses were performed to identify potential predictors of complications. Unadjusted odds ratios (ORs) and 95% confidence intervals (CIs) were calculated. Factors demonstrating statistical significance (*p* < 0.1) in the univariate analysis were then included in a multivariate logistic regression model. Adjusted odds ratios and 95% confidence intervals were derived from the multivariable model. For all analyses, *p*-values less than 0.05 (*p* < 0.05) were considered statistically significant. Missing data were handled using a complete-case analysis approach. Variables with missing values were excluded from the specific analyses in which they were required, and no imputation methods were applied. Given the retrospective nature of the study, missing data were mainly related to variables not systematically recorded in the clinical charts. The proportion of missing data was limited and did not affect the primary endpoint of the study.

## 3. Results

Out of 759 patients who underwent mastectomy, which represented the initial sample, 68 (8.9%) were excluded because they did not undergo breast reconstruction, 41 (5.4%) because they underwent a breast reconstruction with autologous tissue, and 28 (3.7%) due to missing data regarding the type of reconstruction.

A total of 622 patients (81.9%) ultimately underwent mastectomy followed by breast reconstruction with a prothesis implant or tissue expander. Among them, 235 (36.2%) underwent pre-pectoral reconstruction, 366 (56.35%) sub-pectoral, and 21 (3.2%) dual-plane reconstruction, and these were excluded from the analysis. A total of 221 (36.7%) patients underwent direct-to-implant breast reconstruction with a definitive prosthesis, while 380 (63.4%) received reconstruction with a tissue expander. A mesh was used in 60 cases (10.5%) to provide additional support during reconstruction. Among the 601 evaluable patients, 244 (40.6%) underwent conserving mastectomy, 18 (3%) skin-reducing mastectomy, and 339 (55.4%) simplex mastectomy.

The median age was 60 years [range: 28–92 years]. A total of 173 (28.8%) patients underwent neoadjuvant chemotherapy, and 26 (4.4%) patients underwent bilateral mastectomy. The median hospitalization time was 2 days [range: 0–6 days] and the median surgical time was 130 min [range: 60–240 min]. In total, 67 (11.2%) patients were smokers and 22 (3.6%) had type II diabetes mellitus.

Out of the 601 patients evaluated, 29 (4.8%) required a second surgery due to complications and 25 (4.2%) underwent implant removal. An implant infection occurred in 18 (3.0%) patients, post-operative bleeding in 11 (1.8%), breast seroma in 14 (2.3%), breast hematoma in 24 (4.0%), and delayed wound healing in 37 (6.2%).

The median age of patients who underwent pre-pectoral reconstruction was 59.8 [range: 28–92 years], versus 60.4 [range: 34–83 years] in the sub-pectoral group; the relative *p*-value was 0.631. In the pre-pectoral group, 20 patients were smokers, compared with 47 in the sub-pectoral group (*p* = 0.112). Four (1.7%) patients had DMII in the pre-pectoral group and 18 (4.9%) in the sub-pectoral group, showing a statistically significant difference between groups (*p* = 0.045). A lower percentage of patients receiving neoadjuvant chemotherapy was reported in the pre-pectoral group, i.e., 50 (21.3%), compared to 123 (33.6%) in the control group, and the relative *p*-value was 0.002.

The type of mastectomy was statistically different between groups (*p* = 0.029), and the surgical techniques adopted are shown in Table 1. Among the 235 patients who underwent pre-pectoral reconstruction, 112 (47.7%) received DTI reconstruction with a silicon prosthesis, versus 112 (31.1%) in the control group, showing a statistically significant difference. The *p*-values and numbers of patients having undergone breast reconstruction with a tissue expander are reported in Table 1. Regarding breast reconstruction, in the pre-pectoral group, 42 (17.9%) meshes were used in order to provide support, versus 25 (6.9) meshes in the control group, with a *p*-value < 0.001. Indications for mastectomy, the type of axillary surgery performed, and the relative *p*-values are summarized in Table 1. The length of hospitalization was comparable between the different reconstruction techniques adopted, with *p* = 0.964. Differently, the surgical time was significantly shorter in the pre-pectoral group, with a *p*-value < 0.001.

In the pre-pectoral group, 56 (23.8%) patients experienced a high grade of complications, versus 74 (20.2%) patients in the control group, *p* = 0.310 (Table 2). A higher rate of delayed wound healing was reported in the pre-pectoral group at 21 (9.0%), versus 16 (4.3%) patients in the sub-pectoral group, and the relative *p*-value was 0.035. Post-operative bleeding occurred in 11 (4.9%) patients in the sub-pectoral population, while only one case was reported in the pre-pectoral group, *p* = 0.057. Other complications, such as seroma, hematoma, and prothesis infections, were comparable between groups, with *p*-values of 0.793, 0.490, and 0.611, respectively, as shown in Table 2. The rate of implant loss was comparable between the pre-pectoral and sub-pectoral groups, reaching, respectively, 12 (5.1%) vs. 13 (3.6%), *p* = 0.352. Re-operation was required in 14 (6.0%) cases in the pre-pectoral group and in 15 (4.1%) in the control group, *p* = 0.299. Potential delays in adjuvant treatment were reported in 37 (15.7%) cases in the pre-pectoral group versus 49 (13.4%) in the sub-pectoral group; the relative *p*-value was 0.636. No statistically significant differences were observed between the pre-pectoral and sub-pectoral reconstruction groups in terms of overall complications (23.8% vs. 20.2%, *p* = 0.31), minor complications (16.2% vs. 10.9%, *p* = 0.081), or major complications (7.7% vs. 9.3%, *p* = 0.55). Similarly, the overall distribution of post-operative complications (none, minor, and major) did not differ significantly between groups (*p* = 0.09).

In addition, in the multivariate analysis, pre-pectoral reconstruction was not significantly associated with an interval time to adjuvant treatment > 60 days (Table 3). A higher rate of delayed adjuvant treatment, longer than 60 days, was observed in smokers (OR = 2.891, *p* = 0.029), in patients undergoing skin-reducing mastectomy (OR = 2.111, *p* = 0.018), and in those having diabetes (OR = 3.056, *p* = 0.049). There was no statistically significant difference between patients with and without mesh implantation, those undergoing nipple-sparing mastectomy, those older than 60 years old, and patients subjected to ALND.

## 4. Discussion

The number of immediate breast reconstructions has increased in recent years; moreover, more recently, pre-pectoral implants, minimizing surgical trauma, have shown a rapid increase in adoption [14].

Pre-pectoral reconstruction, avoiding muscle dissection, reduces post-operative pain, improves esthetic outcomes, and enables faster functional recovery, allowing patients to resume daily activities more quickly [3,5,6]. On the other hand, as some authors have reported in the literature, the reduced soft-tissue thickness over the prosthesis may increase the risk of complications such as flap ischemia, wound dehiscence, and implant loss [9,10]. In our retrospective single-center study, we observed similar overall complication rates between patients undergoing pre-pectoral and sub-pectoral reconstruction. In the pre-pectoral group, we observed a complication rate of 23.8%, which favorably compares with data reported in the literature, ranging from 20% to 42% [15,16,17]. Since pre-pectoral reconstruction is less invasive, we did not observe a significant increase in complications in this group, likely because the use of meshes was significantly higher. As reported, pre-pectoral implants were associated with mesh use in approximately 20% of cases. In a recent metanalysis, a negative impact on complication rates was reported in patients undergoing breast reconstruction with the use of a mesh [18]. In their analysis of 17 studies, Choi et al. reported an increased risk of implant infection and seroma, but clinically significant complications did not differ between patients who received breast reconstruction with pre-pectoral and sub-pectoral implants [18].

We observed an increased risk of wound dehiscence in patients who underwent pre-pectoral breast reconstruction. This finding is likely related to the greater direct pressure exerted on the mastectomy skin flaps, potentially leading to local ischemic distress and delayed wound healing.

Most of these complications were successfully managed with conservative and outpatient treatments, resulting in minimal discomfort for the patients. Despite the higher incidence of cutaneous complications, the interval to adjuvant treatment was comparable between groups and did not interfere with the timely initiation of oncologic therapies. The relationship between mastectomy skin flap perfusion and ischemic complications remains poorly investigated in the current literature [19]. Reported risk factors include the implant size, body mass index (BMI), smoking status, type II diabetes mellitus, the type of mastectomy, and the incision performed [20,21,22]. The intraoperative assessment of mastectomy flap viability, largely guided by the surgeon’s experience, may influence the decision to use a smaller implant or a partially deflated tissue expander, in order to reduce the mechanical pressure and enhance flap perfusion [19]. With the advent of advanced technologies and the growing integration of artificial intelligence, the objective evaluation of mastectomy flap perfusion should be incorporated into clinical decision-making to reduce post-operative complications, prevent delays in adjuvant therapy, and potentially improve oncologic outcomes [11,12,13,14,15,16,17,18,19,20,21,22,23].

In our series, we observed a higher incidence of post-operative bleeding in patients undergoing sub-pectoral reconstruction. This may be related to the elevation of the highly vascularized pectoralis major muscle [24,25]. Such manipulations can increase the need for blood transfusions or surgical re-interventions, both of which may delay adjuvant treatments and negatively affect oncological outcomes [26]. This procedure is undoubtedly more invasive and has a worse psychological and physical impact on patients. The introduction of pre-pectoral reconstruction has led to an increase in immediate breast reconstructions, significantly improving patients’ quality of life and reducing surgical morbidity, as there is no need to create a sub-pectoral pocket. Although these advantages are well recognized, some authors have reported a higher risk of implant-related complications. Such complications could potentially delay the initiation of adjuvant treatment—a delay that is well known to negatively affect oncologic outcomes. In our study, we compared pre-pectoral and sub-pectoral reconstructions and found no statistically significant differences in the time to adjuvant treatment. Therefore, given its lower surgical impact, the pre-pectoral approach should be considered the preferred option whenever feasible in patients undergoing mastectomy for breast cancer. Moreover, complications did not impact the timing of adjuvant treatment, which remains a key point, as these patients often need to start adjuvant therapies or continue treatments initiated in the neoadjuvant setting.

The operative time was significantly shorter for patients undergoing pre-pectoral reconstruction. This strategy, placing the implant within the plane between the skin and the pre-pectoral fascia, eliminates the additional surgical time required for the pectoralis major muscle’s elevation and the creation of a sub-pectoral pocket. Similar findings have been reported by Sigalove et al., who observed a reduced operative duration in pre-pectoral compared with sub-pectoral reconstruction [26]. In contrast to these findings and to our own, Huang and colleagues reported comparable operative times between the two reconstructive approaches [27]. In our opinion, shortening the surgical time reduces both the duration of anesthesia and the associated immunological stress, as we have reported in some of our previous studies [28,29]. The correlation between immunological impairment and poorer oncological outcomes has been described in the literature and remains a matter of ongoing debate within the scientific community [30,31].

In our study, the implant volume was significantly higher in sub-pectoral reconstructions. This finding is consistent with the previously published literature and may be explained, in our cohort, by the higher proportion of tissue expander-based reconstructions, in which expanders are not fully inflated at the time of the initial surgical procedure [32,33]. However, this difference did not show a significant impact in multivariable logistic regression analyses evaluating risk factors for an interval time to adjuvant treatment greater than 60 days.

In the multivariate analysis, smoking was identified as a significant risk factor for the delayed initiation of adjuvant treatment (OR 2.891). Skin-reducing mastectomy and type II diabetes mellitus were also identified as predictive factors for delays, with odds ratios of 2.111 and 3.056, respectively. These factors are well established and have been consistently reported in the literature by several authors [34,35,36]. All available studies have shown that these patient categories are at a higher risk of post-operative complications. However, no studies in the literature have specifically focused on patients with such comorbidities randomized to different reconstruction types, in order to assess whether pre-pectoral reconstruction should indeed be avoided in this subgroup [37]. For these patients, careful intraoperative assessment of the mastectomy flap thickness and quality is essential, and, in cases of doubt or suspected hypoperfusion, pre-pectoral reconstruction should not be performed. Prospective randomized studies focusing on these populations are warranted to validate our observations.

Recent advances in breast cancer research have highlighted the increasing role of innovative diagnostic and biosensing technologies in improving tissue characterization and treatment personalization. Emerging spectroscopic approaches have demonstrated the potential to provide real-time, non-invasive assessments of the tissue composition and viability, contributing to more accurate intraoperative decision-making [38].

In the field of breast reconstruction, these technological developments parallel the growing interest in the objective intraoperative evaluation of mastectomy skin flap perfusion. Techniques such as indocyanine green (ICG) fluorescence angiography allow the real-time visualization of tissue perfusion, enabling surgeons to better assess flap viability and potentially reduce ischemia-related complications [20,39]. Although perfusion assessment was not routinely performed during the study period and therefore fell beyond the primary scope of the present analysis, the integration of advanced imaging and biosensing technologies may represent an important future direction in optimizing reconstructive outcomes and patient safety.

## 5. Conclusions

According to our analysis, there is no ideal or correct reconstruction type after mastectomy. The reconstructive strategy should be tailored to each patient according to the thickness and quality of the mastectomy flap, while also considering the expected esthetic outcome and existing patient comorbidities. We believe that pre-pectoral reconstruction, being less invasive and associated with shorter operative times, should be preferred whenever feasible, as it has a lower overall impact on the patient. However, this type of reconstruction must be carefully pre-operatively planned by selecting the most appropriate implant, evaluating the potential need for a mesh, and intraoperatively assessing flap perfusion. In cases of documented hypoperfusion or a high risk of complications, the reconstructive plan should be reconsidered, and a sub-pectoral approach may represent a safer alternative to reduce the pressure on the mastectomy flaps.

## 6. Limitations of the Study

This study has several limitations. First, its retrospective design introduces a potential selection bias, as the type of reconstruction was determined by the surgeon’s intraoperative assessment and personal experience, which were not reported or evaluated in the analysis.

The study covered an approximately 10-year period, during which variations may have occurred between breast reconstructions performed in 2014 and those in 2024. These differences likely reflect both the progressive learning curve associated with pre-pectoral breast reconstructions and the technological evolution of the implants and meshes over time.

Given the retrospective nature of the analysis, smoking status was recorded only as a binary variable (smoker/non-smoker), without details regarding the smoking duration, intensity, or its potential effects on microcirculation. Finally, due to the limited number of patients with comorbidities, no subgroup analysis was performed for these specific populations.

## Figures and Tables

**Figure 1 cancers-18-00109-f001:**
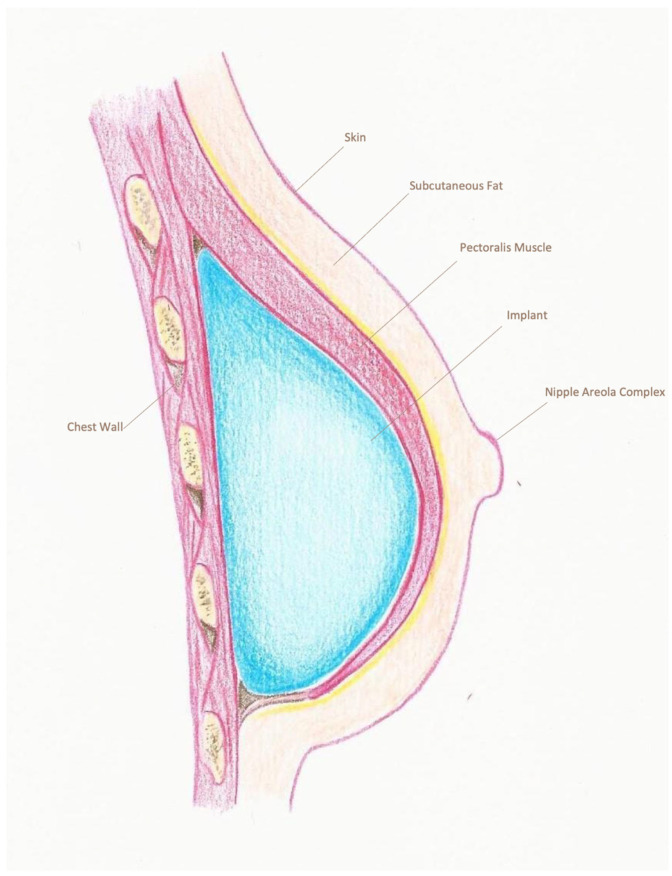
Sub-pectoral breast reconstruction. Lateral view showing implant placement beneath the pectoralis major muscle. The muscle has been elevated to create a sub-pectoral pocket, in which the prosthesis is positioned. This technique provides full muscular coverage of the implant but requires detachment of the pectoralis major, which may increase post-operative pain and recovery time.

**Figure 2 cancers-18-00109-f002:**
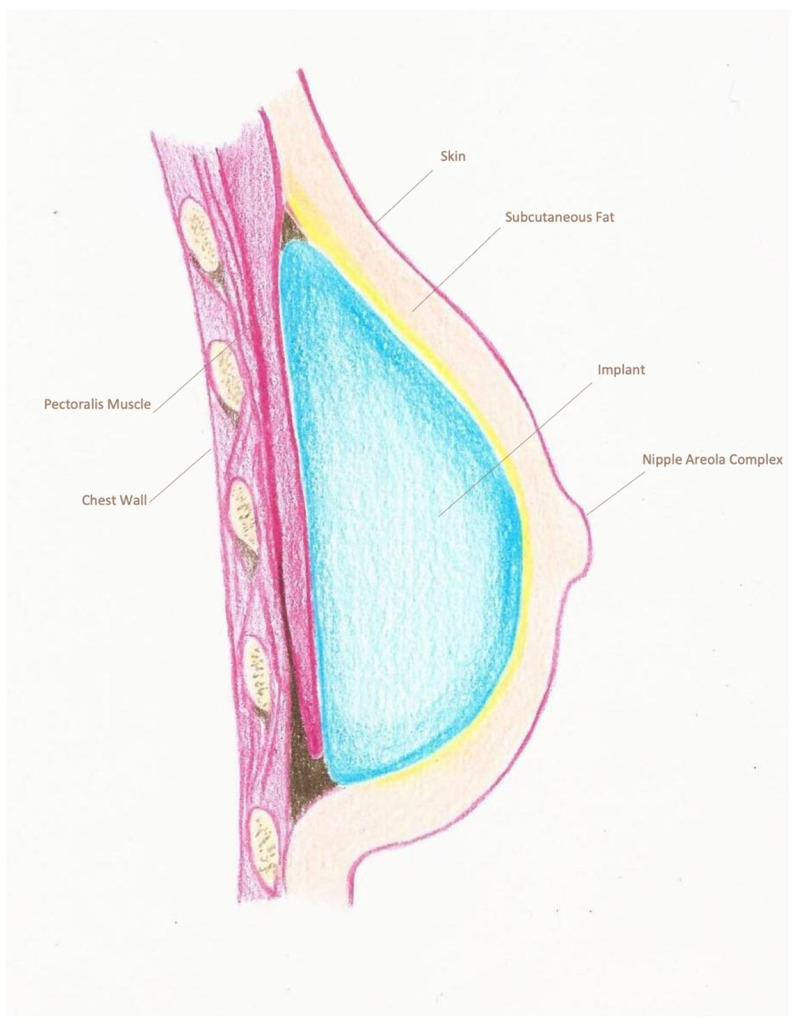
Pre-pectoral breast reconstruction without mesh. The implant is positioned directly over the pectoralis major muscle, between the mastectomy skin flap and the pectoral fascia, without the use of a mesh or acellular dermal matrix. This approach avoids muscular dissection, resulting in reduced surgical time and post-operative discomfort.

**Figure 3 cancers-18-00109-f003:**
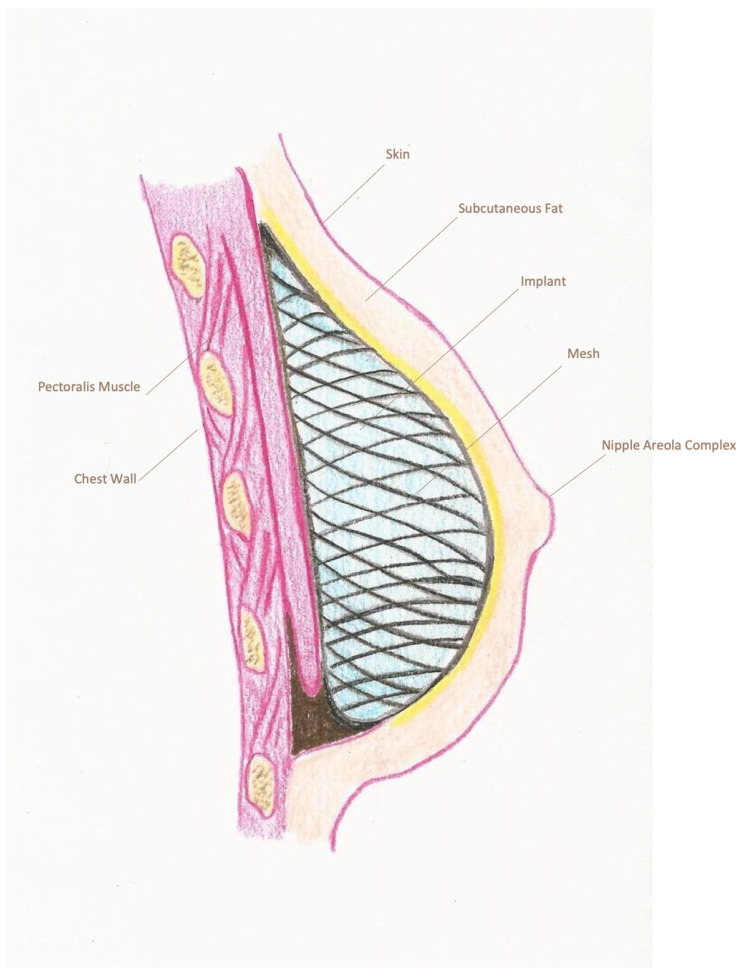
Pre-pectoral breast reconstruction with mesh. Pre-pectoral implant-based reconstruction using complete mesh coverage. The implant is wrapped with an acellular dermal matrix (ADM), providing additional support and improving contour definition. The mesh helps to reduce implant visibility and rippling while enhancing soft-tissue coverage and overall esthetic outcomes.

**Table 1 cancers-18-00109-t001:** Surgical procedures.

	Pre-Pectoral Group (*n* = 235)	Sub-Pectoral Group (*n* = 366)	*p*-Value
**Age (years)**	59.8 [28–92]	60.4 [34–83]	0.631
**BMI (kg/m^2^)**	23.2 [20.1–33.2]	23.7 [20.9–34.1]	0.882
**Smoker (Yes)**	33 (14.5%)	34 (12.6%)	0.112
**Diabetes Mellitus II (Yes)**	4 (1.7%)	18 (4.9%)	0.045
**Bilateral Mastectomy (Yes)**	45 (19.2%)	78 (21.4%)	0.533
**Neoadjuvant CHT (Yes)**	50 (21.3%)	123 (33.6%)	0.002
**Previous RT**	13 (5.5%)	23 (6.3%)	0.860
*>10 Years Before*	9 (3.8%)	13 (3.5%)	1.000
*<10 Years Before*	4 (1.7%)	10 (2.8%)	0.581
**Radiological MSF Thickness**			0.075
*Type I*	51 (21.7%)	109 (29.8%)	
*Type II*	149 (63.4%)	205 (56.1%)	
*Type III*	38 (16.2%)	52 (14.2%)	
**Type of Mastectomy**			0.029
*Simplex*	146 (62.1%)	193 (52.7%)	
*Skin Sparing*	21 (8.9%)	64 (17.5%)	
*NAC Sparing*	62 (26.4%)	97 (26.5%)	
*Skin Reducing*	6 (2.5%)	12 (3.3%)	
**Type of Implant**			<0.001
*DTI*	112 (47.7%)	114 (31.1%)	
*Tissue Expander*	123 (52.3%)	252 (68.9%)	
**Mesh (Yes)**	42 (17.9%)	25 (6.9%)	<0.001
**Mastectomy Indications**			0.445
*Invasive Carcinoma **	187 (79.5%)	304 (83.1%)	
*Pure DCIS*	28 (11.9%)	40 (10.9%)	
*Other ***	20 (7.5%)	22 (6.1%)	
**Specimen Breast Volume (cc)**	476.3 ± 193.4	501.1 ± 211.3	0.141
**Implant Breast Volume (cc)**	423.3 ± 81.7	451.5 ± 96.0	0.001
**Axillary Surgery**			0.944
*SNLB*	124 (52.8%)	199 (54.4%)	
*ALND*	76 (32.3%)	123 (33.6%)	
*NO*	25 (10.6%)	44 (12.1%)	
**Adjuvant RT**	10 (4.3%)	19 (5.2%)	0.698
**Hospitalization**	2 [0:5]	2 [1:6]	0.964
**Surgical Time**	135 [60:167]	141 [89:240]	<0.001

BMI: body mass index; CHT: chemotherapy; RT: radiation therapy; NAC: nipple–areola complex; MSF: mastectomy skin flap; DTI: direct to implant; * ductal or lobular carcinoma; DCIS: ductal carcinoma in situ; ** mixed type or other type; SNLB: sentinel lymph node biopsy; ALND: axillary lymph node dissection; hospitalization in days; surgical time in minutes.

**Table 2 cancers-18-00109-t002:** Complications and surgical outcomes.

	Pre-Pectoral Group (*n* = 235)	Sub-Pectoral Group (*n* = 366)	*p*-Value
**C** **omplications**	56 (23.8%)	74 (20.2%)	0.310
**Minor Complications (I–II)**	38 (16.2%)	40 (10.9%)	0.081
**Major Complications (IIIa–IIIb)**	18 (7.7%)	34 (9.3%)	0.55
**D** **elayed Wound Healing**	21 (9.0%)	16 (4.3%)	0.035
**S** **eroma**	5 (2.1%)	9 (2.5%)	0.793
**H** **ematoma**	11 (4.7%)	13 (3.5%)	0.490
**B** **leeding**	1 (1.7%)	11 (4.9%)	0.057
**I** **mplant Infection**	6 (2.6%)	12 (3.3%)	0.611
**I** **mplant Loss**	12 (5.1%)	13 (3.6%)	0.352
**R** **e-Operation**	14 (6.0%)	15 (4.1%)	0.299
**I** **nterval Time to Adjuvant Treatment > 60 days**	37 (15.7%)	49 (13.4%)	0.636

Complications according to the Clavien–Dindo Classification.

**Table 3 cancers-18-00109-t003:** Multivariate logistic regression analyses to identify risk factors for interval time to adjuvant treatment > 60 days.

	OR	95% CI	*p*-Value
**Implant Position Pre-Pectoral**	0.912	0.511–1.321	0.652
**Mesh (Yes)**	0.655	0.210–1.711	0.115
**Smoker (Yes)**	2.891	0.915–3.716	0.029
**Diabetes M II (Yes)**	3.056	1.201–8.612	0.049
**Skin-Reducing Mastectomy**	2.111	1.337–4.391	0.018
**Nipple-Sparing Mastectomy**	0.774	0.235–1.661	0.092
**Simple Mastectomy**	0.341	0.122–0.921	0.133
**Age (>60 years old)**	1.136	0.647–1.953	0.602
**ALND (Yes)**	1.219	0.407–3.524	0.391
***Implant Breast Volume*** **(>400 cc)**	1.078	0.335–1.869	0.444

## Data Availability

The data presented in this study are available upon request from the corresponding author, subject to valid justification.

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
