# Peer review of "Post-Operative Complications Do Not Influence Time to Adjuvant Treatment in Breast Cancer Patients Undergoing Implant-Based Reconstructions: Pre-Pectoral Versus Sub-Pectoral"

_cancers, 2025, doi:10.3390/cancers18010109_

Round 1
Reviewer 1 Report
Comments and Suggestions for Authors
The topic seems to be interesting but what is the rationale of your research or the experimental approach that you are taking, and what are the new fundamental understanding or the technical innovation of the work in the context of research reported in literature. Statistical data analysis is also missing.
In another word, the manuscript needs to identify what is the true selling point or what the readers can learn from it in the sense of either fundamentally or technically. In general, having done some experiments and acquiring some technical data will not make a sound publication.
I reject this paper.
Author Response
Thank you for pointing this out; we have made an effort to improve our work accordingly. The majority of studies on pre-pectoral reconstruction focus primarily on surgical outcomes or postoperative complications. Our article offers an innovative perspective by adopting an oncological viewpoint, specifically evaluating the impact of reconstruction techniques on the timing of adjuvant treatments, which is a critical factor for breast cancer patients. Given that pre-pectoral reconstruction appears to be associated with reduced surgical impact and lower oncological stress, we believe it should be considered whenever feasible in this patient population.
Reviewer 2 Report
Comments and Suggestions for Authors
Report of Manuscript (ID cancers-4011126) titled “Post-operative complications do not influence time to adjuvant treatment in breast cancer patients undergoing implant-based reconstructions: Pre-Pectoral versus Sub-Pectoral”
Overall Recommendation: Moderate Revision
Thank you for the opportunity to review this manuscript. The study addresses an important and timely question in breast cancer surgery. The comparison between prepectoral and subpectoral implant-based reconstruction and their potential impact on the initiation of adjuvant therapy is clinically relevant, and the large sample size represents a notable strength. The manuscript is generally well written, and the conclusions are consistent with the data presented.
However, before this work can be considered for publication in a high-impact journal, some methodological, analytical, and reporting issues should be addressed. These relate primarily to selection bias, baseline imbalances, limitations of the statistical approach, and the need for clearer justification and transparency regarding key methodological decisions.
I kindly ask the authors to carefully address the following major and minor comments.
Major Comments
(a) This retrospective study relies on intraoperative surgeon judgment to determine the type of reconstruction. This approach carries a substantial risk of selection bias. In addition, several baseline characteristics differ significantly between groups (e.g., neoadjuvant chemotherapy, diabetes, DTI rate, mastectomy type, mesh use), potentially confounding the relationship between reconstruction technique and postoperative outcomes.
Please consider performing a propensity score analysis (matching or inverse probability weighting) or, at least, a more comprehensive adjusted regression model that includes all significantly imbalanced variables. This would substantially strengthen the validity of the conclusions.
(b) The use of a binary cutoff (>60 days) for delayed adjuvant therapy reduces a continuous or time-to-event outcome to a dichotomous measure. The rationale for selecting 60 days is not clearly explained. Please clarify the choice of this threshold and consider providing time-to-event analyses (e.g., Kaplan–Meier curves, Cox regression), if feasible. These analyses would offer a more robust assessment of potential differences between groups.
(c) Although the cohort is large, several comparisons approach statistical significance, suggesting possible type II error. Please include a post-hoc power analysis or discuss the study’s power to detect clinically meaningful differences in key outcomes.
(d) The manuscript states that reconstruction type was chosen based on intraoperative assessment, but the criteria guiding this choice are not described. This information is essential for contextualizing potential selection bias. Could the authors please clarify:
-What specific factors guided the choice between prepectoral and subpectoral reconstruction?
-Were flap thickness, flap perfusion, patient comorbidities, or other intraoperative findings part of the decision-making process?
-Did practice patterns evolve between 2014 and 2024, and if so, how?
A clearer explanation is needed to understand the comparability between the two groups.
(e) The current figures are primarily descriptive surgical illustrations and do not contribute meaningfully to the analytic results.
Please consider removing or replacing them with more informative visualizations, such as a flowchart showing patient selection and exclusions, distribution of complications between groups, or time-to-adjuvant-therapy curves.
(f) Complications are reported only as binary outcomes. Including a severity scale (e.g., Clavien–Dindo or a similar classification) would enhance the reproducibility and interpretability of the findings. If possible, please include a complication severity grading system. (Not mandatory but only desirable now or for future further investigation)
Minor Comments
- Please adopt a neutral, evidence-based tone and reduce the use of subjective expressions such as “in our opinion.”
- Some sentences would benefit from stylistic and grammatical refinement; a thorough language revision is recommended.
- Please describe how missing data were handled (e.g., complete-case analysis, imputation).
- In the Tables, please indicate statistically significant differences with asterisks; ensure all units and ranges are consistently reported; use consistent statistical formatting (e.g., p < 0.05).
- Consider streamlining the reference list to remove redundancies.
Grammar and Style Suggestions
Use consistent American English spelling (e.g., “anesthesia”, “tumor”, “behavior”, “hemorrhage”).
Add missing definite articles: “the implant” instead of “implant” - “the pectoralis major muscle” instead of “pectoralis major muscle”.
Replace “worst psychological and physical impact” → “greater psychological and physical impact”
Replace “prothesis infection” → “implant infection”
Replace “mesh were used” → “mesh was used”
Remove doubled wording: “with or without or without mesh coverage” → “with or without mesh coverage”
Ensure consistent statistical formatting: use p < 0.05 (with spaces), use p-values (with a hyphen)
Avoid comma splices in long sentences.
Use correct plural forms:
“odds ratios (ORs)” instead of “odds ratio (OR)”
“confidence intervals (CIs)” instead of “confidence interval (CI)”
“complications, such as flap ischemia…” (no colon before such as)
“recoreds” → “records”
“vallidate” → “validate”
Reduce subjective expressions; replace:
“in our opinion, given its lower surgical impact…” → “Given its lower surgical impact, the prepectoral approach may be preferred when feasible.”
Replace “in our series we observed” with more neutral phrasing “In this cohort, we observed…”
Prefer neutral scientific phrasing.
Final Assessment
The study is promising and clinically relevant. With substantial methodological refinement and clearer reporting, it could represent a valuable contribution to the field of breast reconstruction and oncologic outcomes. Complimenti ed in bocca al lupo per il lavoro fatto ragazzi !!!
Thank you once again for the opportunity to review this manuscript. I hope the comments above will assist the authors in strengthening their work.
Author Response
Dear Reviewer, thank you for your time.
We sincerely thank you for giving us the opportunity to improve our work through your valuable suggestions. We have addressed each of your comments point by point, and the corresponding revisions have been clearly highlighted in the revised manuscript. We remain at your disposal for any further clarifications or additional requests that may help improve the quality of our manuscript.
Kinds regards
Marco Pellicciaro
Overall Recommendation: Moderate Revision
Thank you for the opportunity to review this manuscript. The study addresses an important and timely question in breast cancer surgery. The comparison between prepectoral and subpectoral implant-based reconstruction and their potential impact on the initiation of adjuvant therapy is clinically relevant, and the large sample size represents a notable strength. The manuscript is generally well written, and the conclusions are consistent with the data presented.
However, before this work can be considered for publication in a high-impact journal, some methodological, analytical, and reporting issues should be addressed. These relate primarily to selection bias, baseline imbalances, limitations of the statistical approach, and the need for clearer justification and transparency regarding key methodological decisions.
I kindly ask the authors to carefully address the following major and minor comments.
Major Comments
- This retrospective study relies on intraoperative surgeon judgment to determine the type of reconstruction. This approach carries a substantial risk of selection bias. In addition, several baseline characteristics differ significantly between groups (e.g., neoadjuvant chemotherapy, diabetes, DTI rate, mastectomy type, mesh use), potentially confounding the relationship between reconstruction technique and postoperative outcomes. Please consider performing a propensity score analysis (matching or inverse probability weighting) or, at least, a more comprehensive adjusted regression model that includes all significantly imbalanced variables. This would substantially strengthen the validity of the conclusions.
We attempted to mitigate this bias by including additional baseline characteristics in the analysis. Unfortunately, due to the retrospective nature of the study and routine clinical practice in previous years, higher-risk patients were more often reconstructed using the subpectoral approach. While we acknowledge the presence of potential selection biases, which could be more appropriately addressed in future prospective studies, their impact was minimized through multivariable analyses aimed at evaluating factors influencing delays in adjuvant treatment, which represents the primary objective of our study.
Although propensity score, based methods may further reduce imbalance, their application in our cohort would have resulted in a substantial reduction of the effective sample size and statistical power, particularly given the relatively limited number of events. For this reason, we considered a multivariable logistic regression model to be the most appropriate and robust approach for the aims of the present study. As suggested, we have implemented logistic regression analysis.
(b) The use of a binary cutoff (>60 days) for delayed adjuvant therapy reduces a continuous or time-to-event outcome to a dichotomous measure. The rationale for selecting 60 days is not clearly explained. Please clarify the choice of this threshold and consider providing time-to-event analyses (e.g., Kaplan–Meier curves, Cox regression), if feasible. These analyses would offer a more robust assessment of potential differences between groups.
The >60-day cutoff was selected based on an internal institutional parameter, reflecting the standard timeframe within which patients at our center typically initiate adjuvant treatment in routine clinical practice. We have clarified this rationale in the Materials and Methods section of the revised manuscript. Unfortunately, due to the retrospective nature of the study, detailed time-to-event data were not consistently available for all patients, which precluded the performance of time-to-event analyses such as Kaplan–Meier curves or Cox regression. We appreciate this valuable suggestion and would like to emphasize that, in the prospective study we are currently conducting, we plan to analyze the timing of adjuvant treatment initiation in greater detail, including appropriate time-to-event analyses. (c) Although the cohort is large, several comparisons approach statistical significance, suggesting possible type II error. Please include a post-hoc power analysis or discuss the study’s power to detect clinically meaningful differences in key outcomes.
(d) The manuscript states that reconstruction type was chosen based on intraoperative assessment, but the criteria guiding this choice are not described. This information is essential for contextualizing potential selection bias. Could the authors please clarify:
-What specific factors guided the choice between prepectoral and subpectoral reconstruction?
-Were flap thickness, flap perfusion, patient comorbidities, or other intraoperative findings part of the decision-making process?
-Did practice patterns evolve between 2014 and 2024, and if so, how?
A clearer explanation is needed to understand the comparability between the two groups.
Unfortunately, due to the retrospective nature of the study, detailed information regarding the specific rationale for the surgical choice was not always available, as it was not systematically reported in the operative records. However, the decision-making process was consistent with routine clinical practice and was primarily guided by intraoperative assessment, including mastectomy flap characteristics and thickness. There were no substantial changes in institutional policy at our center during the study period, apart from a progressively increased confidence in the use of the pre-pectoral reconstruction technique
(e) The current figures are primarily descriptive surgical illustrations and do not contribute meaningfully to the analytic results.
Please consider removing or replacing them with more informative visualizations, such as a flowchart showing patient selection and exclusions, distribution of complications between groups, or time-to-adjuvant-therapy curves.
We believe it would be useful to add illustrative figures, as this may help readers—particularly oncologists who are less familiar with reconstructive techniques—to better understand the surgical approaches and their potential correlation with delays in adjuvant treatments. We were not able to perform time-to-event analyses or survival curves because the necessary continuous timing data were not consistently available, given the retrospective nature of the study.
(f) Complications are reported only as binary outcomes. Including a severity scale (e.g., Clavien–Dindo or a similar classification) would enhance the reproducibility and interpretability of the findings. If possible, please include a complication severity grading system. (Not mandatory but only desirable now or for future further investigation)
We have included a complication severity grading system, as you requested.
Minor Comments
- Please adopt a neutral, evidence-based tone and reduce the use of subjective expressions such as “in our opinion.”
As suggested, we have avoided the use of subjective expressions. Thank you for drawing our attention to this point.
- Some sentences would benefit from stylistic and grammatical refinement; a thorough language revision is recommended.
We have performed an English language revision; however, as agreed with the Editor, should the manuscript be accepted, we will proceed with a further English revision using the MDPI editing service. This arrangement was made for administrative and funding-related reasons, as it is easier for us to use departmental funds after acceptance.
- Please describe how missing data were handled (e.g., complete-case analysis, imputation).
Thank you for this comment. Missing data were handled using a complete-case analysis approach. Variables with missing values were excluded from the specific analyses in which they were required, and no imputation methods were applied. Given the retrospective nature of the study, missing data were mainly related to variables not systematically recorded in the clinical charts. The proportion of missing data was limited and did not affect the primary endpoint of the study. This aspect has now been clearly specified in the Materials and Methods section.
- In the Tables, please indicate statistically significant differences with asterisks; ensure all units and ranges are consistently reported; use consistent statistical formatting (e.g., p < 0.05).
Thank you for pointing this out. We also agree that highlighting statistically significant differences can improve the readability of the tables. In a previous manuscript, we indicated p values < 0.05 in bold; however, during the editorial process, the MDPI editors requested the removal of this formatting, considering statistical significance to be an objective parameter that does not require graphical emphasis. Regarding the use of thresholds such as p < 0.05, given that our sample size is not extremely large and that some events, particularly in the multivariable analysis, are relatively infrequent, we preferred to report the exact p values to provide a more precise and transparent interpretation of the results.
- Consider streamlining the reference list to remove redundancies.
We carefully reviewed the reference list and removed redundant citations. In particular, duplicate references addressing the same study were consolidated, and the reference list was streamlined accordingly.
Grammar and Style Suggestions
Use consistent American English spelling (e.g., “anesthesia”, “tumor”, “behavior”, “hemorrhage”).
Thank you for this suggestion. We have revised the manuscript to ensure consistent use of American English spelling throughout (e.g., “anesthesia”, “tumor”, “behavior”, “hemorrhage”). A further language improvement will be provided through the MDPI English editing service should the manuscript be accepted.
Add missing definite articles: “the implant” instead of “implant” - “the pectoralis major muscle” instead of “pectoralis major muscle”.
We have addressed this comment by adding the missing definite articles throughout the manuscript (e.g., “the implant,” “the pectoralis major muscle”) to ensure grammatical consistency and clarity.
Reviewer 3 Report
Comments and Suggestions for Authors
The manuscript addresses an important and clinically relevant question; however, several methodological and reporting limitations weaken the strength of the conclusions. Following comments need to be addressed,
@ The introduction misses articulation of the research gap and the specific problem which needs to be addressed.
@ The manuscript does not adequately control for baseline clinical differences such as diabetes, smoking status, and neoadjuvant chemotherapy, which may independently influence complication rates and confound the reported associations.
@ Radiotherapy parameters such as dose, fractionation, sequencing, and timing relative to reconstruction are not described, limiting the ability to contextualize the complication rates reported in irradiated patients.
@ The surgical technique is not addressed properly, particularly regarding mastectomy flap thickness assessment and intraoperative perfusion evaluation, both of which strongly influence postoperative complications.
@ There is no stratified analysis comparing direct-to-implant versus tissue expander reconstruction, which limits the clarity of how each modality independently contributed to the overall complication profile.
@ Key implant characteristics such as volume, fill type, placement plane, and surface texture are not reported, making it difficult to assess their contribution to outcomes.
@ The study does not classify complications by severity grade or management approach, preventing a nuanced interpretation of clinical relevance and impact on patient recovery.
@ Important perioperative variables such as operative time, use of electrocautery, and specimen weight are missing, all of which can significantly impact complication risk.
@ No assessment is provided on patient-related factors such as BMI categories or comorbidity indices, preventing readers from evaluating risk-adjusted complication patterns.
@ The literature review would benefit from integrating several recent and highly relevant studies that address advanced diagnostic, therapeutic, and molecular characterization approaches in breast cancer. Incorporating works such as Zeng et al. (2023, doi: 10.1016/j.saa.2022.122000) and Ma et al. (2020, doi: 10.3788/COL202018.051701) would strengthen the context for emerging spectroscopic and biosensing technologies. Similarly, innovative therapeutic strategies and biomarker discovery frameworks described by Yuan et al. (2025, doi: 10.1007/s40005-025-00731-z) and Li et al. (2021, doi: 10.1109/TCBB.2020.2973148) provide valuable insights that the manuscript currently lacks. Additional inclusion of recent immunological and mechanistic studies—such as Wang et al. (2024, doi: 10.1111/imm.13793), Yang et al. (2025, doi: 10.1016/j.vaccine.2024.126635), and Han et al. (2025, doi: 10.1016/j.redox.2025.103843) would further enhance the scientific depth by offering updated perspectives on tumor microenvironment interactions, treatment-associated risks, and broader clinical implications. Integrating these references will create a more comprehensive and current scholarly foundation for the study.
Author Response
Dear Reviewer, thank you for your time.
We sincerely thank you for giving us the opportunity to improve our work through your valuable suggestions. We have addressed each of your comments point by point, and the corresponding revisions have been clearly highlighted in the revised manuscript. We remain at your disposal for any further clarifications or additional requests that may help improve the quality of our manuscript.
Kinds regards
Marco Pellicciaro
The manuscript addresses an important and clinically relevant question; however, several methodological and reporting limitations weaken the strength of the conclusions. Following comments need to be addressed,
@ The introduction misses articulation of the research gap and the specific problem which needs to be addressed.
Thank you. We have expanded the Introduction in line with the reviewers’ suggestions, and we hope that the research gap addressed by this study is now more clearly articulated.
@ The manuscript does not adequately control for baseline clinical differences such as diabetes, smoking status, and neoadjuvant chemotherapy, which may independently influence complication rates and confound the reported associations.
All these baseline characteristics were reported both in the tables and in the Results section, as we consider them to play an important role in influencing complication rates. To further clarify this aspect and avoid any possible misunderstanding, the manuscript has been revised accordingly. Among these factors, only type II diabetes mellitus and prior neoadjuvant therapy showed a statistically significant difference between the two groups. As is well known, these patients are generally considered at higher risk, and for this reason subpectoral reconstruction was more frequently preferred according to institutional policy. This potential selection bias has been acknowledged and discussed in the Limitations section. Moreover, these variables were included in the multivariable analysis to assess their independent impact on the risk of delay in adjuvant treatments, which represents a key outcome for oncologic patients. We agree that this limitation will be addressed in the prospective randomized study that we are currently initiating.
@ Radiotherapy parameters such as dose, fractionation, sequencing, and timing relative to reconstruction are not described, limiting the ability to contextualize the complication rates reported in irradiated patients.
Thank you for pointing out this omission. We have added to Table 1 the data regarding patients who underwent preoperative radiotherapy, including the relative timing, without observing statistically significant differences between the groups. We have also included the cases of adjuvant radiotherapy in the revised manuscript. Unfortunately, given the retrospective nature of the study, we were not able to retrieve detailed radiotherapy parameters such as dose, fractionation, sequencing, and timing for these patients. However, according to institutional policy, adjuvant radiotherapy is routinely administered within 3 months after surgery. We have accordingly revised both the Materials and Methods and the Results sections.
@ The surgical technique is not addressed properly, particularly regarding mastectomy flap thickness assessment and intraoperative perfusion evaluation, both of which strongly influence postoperative complications.
During the study period (2014–2024), intraoperative assessment of mastectomy flap perfusion was not routinely performed in our institution, in line with the practice of the majority of centers in Italy at that time. More recently, we have initiated a prospective observational study specifically designed to evaluate flap perfusion using indocyanine green angiography, flap thickness, and postoperative complications. However, this analysis falls outside the scope of the present study. Given the retrospective nature of the study, we were not able to retrieve intraoperative flap thickness measurements expressed in millimeters. However, through a detailed review of the available data, flap thickness was classified according to the Rancati classification, which has been reported in the revised manuscript. The primary objective of our work was to assess whether pre-pectoral breast reconstruction has an impact on the timing of adjuvant treatments, rather than to investigate technical intraoperative factors influencing postoperative complications. We have accordingly revised both the Materials and Methods and the Results sections.
@ There is no stratified analysis comparing direct-to-implant versus tissue expander reconstruction, which limits the clarity of how each modality independently contributed to the overall complication profile.
@ Key implant characteristics such as volume, fill type, placement plane, and surface texture are not reported, making it difficult to assess their contribution to outcomes.
Regarding the placement plane, this analysis specifically compares pre-pectoral versus subpectoral reconstructions, explicitly excluding dual-plane techniques; therefore, we believe this aspect is clearly defined. Thank you for prompting us to further consider this point. We have added the mean volume of the implants used. With respect to implant type, according to our institutional policy, only anatomical breast implants are used for reconstruction. The implant volume was higher in patients undergoing mastectomy with pre-pectoral reconstruction. However, this difference did not show a significant impact in multivariable logistic regression analyses evaluating risk factors for an interval time to adjuvant treatment greater than 60 days. Throughout the study period, silicone implants with a microtextured surface were consistently used for breast reconstruction.
@ The study does not classify complications by severity grade or management approach, preventing a nuanced interpretation of clinical relevance and impact on patient recovery.
We initially did not include a formal classification of complications to avoid excessive fragmentation of the results and to maintain the focus on the type of complications and their potential impact on delays in adjuvant treatment. However, in response to the reviewer’s comment, we have now incorporated the Clavien–Dindo classification to better stratify the clinical relevance of postoperative complications. This classification has been added to the Results tables, and a corresponding addendum has been included in the Materials and Methods section.
@ Important perioperative variables such as operative time, use of electrocautery, and specimen weight are missing, all of which can significantly impact complication risk.
Operative time was reported in the Results section and analyzed, showing a reduction in operative time in patients undergoing pre-pectoral reconstruction. This difference can be explained by the fact that, in subpectoral reconstructions, creation of the pocket behind the pectoralis major muscle is required, which is a more time-consuming step. This aspect has been analyzed and discussed in the manuscript. Regarding the use of electrocautery, we have clarified in the Materials and Methods section how mastectomy is performed at our institution. According to our internal policy, electrocautery is avoided in the superficial plane in order to minimize thermal injury to the skin flaps. Although the weight of the excised breast specimens was not available, we have included the volume of the surgical specimens, which we believe provides a more informative measure of breast size. We hope these clarifications adequately address the reviewer’s concern.
@ No assessment is provided on patient-related factors such as BMI categories or comorbidity indices, preventing readers from evaluating risk-adjusted complication patterns.
BMI did not differ significantly between the two groups and was not associated with postoperative complications; therefore, it was not included in the multivariable analysis. Comorbidities were included in the multivariable model and did not show a significant impact on the risk of an interval time to adjuvant treatment greater than 60 days. Performing additional subgroup analyses would not be statistically meaningful, as the relatively small number of patients with comorbidities would not allow for a robust multivariable analysis and would provide limited scientific relevance.
@ The literature review would benefit from integrating several recent and highly relevant studies that address advanced diagnostic, therapeutic, and molecular characterization approaches in breast cancer. Incorporating works such as Zeng et al. (2023, doi: 10.1016/j.saa.2022.122000) and Ma et al. (2020, doi: 10.3788/COL202018.051701) would strengthen the context for emerging spectroscopic and biosensing technologies. Similarly, innovative therapeutic strategies and biomarker discovery frameworks described by Yuan et al. (2025, doi: 10.1007/s40005-025-00731-z) and Li et al. (2021, doi: 10.1109/TCBB.2020.2973148) provide valuable insights that the manuscript currently lacks. Additional inclusion of recent immunological and mechanistic studies—such as Wang et al. (2024, doi: 10.1111/imm.13793), Yang et al. (2025, doi: 10.1016/j.vaccine.2024.126635), and Han et al. (2025, doi: 10.1016/j.redox.2025.103843) would further enhance the scientific depth by offering updated perspectives on tumor microenvironment interactions, treatment-associated risks, and broader clinical implications. Integrating these references will create a more comprehensive and current scholarly foundation for the study.
Thank you for the suggestion to expand the Discussion. We have addressed this comment by including a brief discussion of emerging technologies for intraoperative perfusion assessment, which are the most relevant to the surgical and reconstructive focus of the present study, although not its primary aim. The additional references suggested on nano-assembly of chemotherapeutic agents, biomarker-based survival prediction, and immunological or mechanistic aspects were carefully considered; however, we believe that these topics fall outside the scope of this manuscript, which is centered on surgical reconstruction and clinical outcomes. To maintain scientific coherence and appropriateness of citations, these references were therefore not included.
Round 2
Reviewer 2 Report
Comments and Suggestions for Authors
Dear All, thank you for reviewing the manuscript and incorporating my suggestions. The manuscript continues to be valid and well organized. The improvements have made the manuscript more solid. I recommend publication.
Best regards
Reviewer 3 Report
Comments and Suggestions for Authors
The comments are addressed.